# Evaluation of Anti-PT Antibody Response after Pertussis Vaccination and Infection: The Importance of Both Quantity and Quality

**DOI:** 10.3390/toxins13080508

**Published:** 2021-07-21

**Authors:** Alex-Mikael Barkoff, Aapo Knuutila, Jussi Mertsola, Qiushui He

**Affiliations:** 1Research Center for Infection and Immunity, Institute of Biomedicine, University of Turku, 20520 Turku, Finland; ambark@utu.fi (A.-M.B.); aajukn@utu.fi (A.K.); jusmer@utu.fi (J.M.); 2Department of Paediatrics and Adolescent Medicine, Turku University Hospital, 20520 Turku, Finland; 3InFLAMES Research Flagship Center, University of Turku, 20520 Turku, Finland

**Keywords:** antibodies, avidity, B cells, CHO cell, epitopes, pertussis toxin

## Abstract

Pertussis toxin (PT) is considered the main virulence factor causing whooping cough or pertussis. The protein is widely studied and its composition was revealed and sequenced already during the 1980s. The human immune system creates a good response against PT when measured in quantity. However, the serum anti-PT antibodies wane rapidly, and only a small amount of these antibodies are found a few years after vaccination/infection. Therefore, multiple approaches to study the functionality (quality) of these antibodies, e.g., avidity, neutralizing capacity, and epitope specificity, have been investigated. In addition, the long-term B cell memory (Bmem) to PT is crucial for good protection throughout life. In this review, we summarize the findings from functional PT antibody and Bmem studies. These results are discussed in line with the quantity of serum anti-PT antibodies. PT neutralizing antibodies and anti-PT antibodies with proper avidity are crucial for good protection against the disease, and certain epitopes have been identified to have multiple functions in the protection. Although PT-specific Bmem responses are detectable at least five years after vaccination, long-term surveillance is lacking. Variation of the natural boosting of circulating *Bordetella pertussis* in communities is an important confounding factor in these memory studies.

## 1. Introduction

*Bordetella pertussis* produces several virulence factors. These could be classified into two main groups as adhesins and toxins. Several of the adhesins are components of the current acellular pertussis vaccines (ACV) in various compositions. These are filamentous hemagglutinin (FHA), pertactin (PRN), and fimbriae (FIM). Most of the toxins are proteins such as pertussis toxin (PT), adenylate cyclase toxin, and dermonecrotic toxin. Non-protein toxins are tracheal cytotoxin, which is a fragment of the peptidoglycan, and lipo-oligosaccharide (endotoxin). Detoxified PT is the major component of all current ACVs.

PT is a major virulence factor of *B. pertussis* [1] and is a combination of five subunits (S1–5) (Figure 1). S1 has ADP-ribosyltransferase activity using NAD as ADP-ribosyl donor and signal transduction, G proteins as ADP-ribosyl acceptors [2]. PT is secreted through the *B. pertussis* cell membrane to its surroundings, and as a consequence of the PT actions in the intracellular processes, there will be a regulatory imbalance such as the uncontrolled formation of cAMP, which will lead to metabolic changes and paralysis in the target cells. The S2–5 binds to various (but mostly unidentified) glycoconjugate molecules on the surfaces of eukaryotic cells, and is involved in the translocation of the toxic S1 across the cell membrane [3]. Historically, before the identification of actions of just one protein, PT had several names that reflect its different toxic properties: islet-activating protein, histamine-sensitizing factor, and lymphocytosis-promoting factor. Additionally, PT is also a mitogen and an adjuvant [4].

In 1974, Yuji Sato and Hiroko Sato reported the isolation and characterization of protective antigens of *B. pertussis* tested in mice [5]. This led to the development and use of ACVs, which mainly contained formalin detoxified PT and FHA. ACVs have been used in Japan since 1981. One of the main advantages of ACVs, as compared to whole-cell vaccines (WCV), is the removal of endotoxin, which is the main cause of fever and other side effects related to earlier vaccines. Locht et al., first described, cloned, and sequenced the PT gene, which was later confirmed by the group of Rino Rappuoli [4,6]. A new genetically inactivated recombinant PT vaccine was developed, in which the enzymatic activity was inactivated through precise genetic modifications. Later, the recombinant PT vaccine was combined with FHA and PRN and was shown to be immunogenic and safe in infants, and induced early and long-lasting protection [7,8,9].

One problem in the understanding of pertussis pathophysiology is the lack of biomarkers, which indicate protection against the disease. Hemagglutinating antigens (PT and FHA) are important virulence factors [5]. Historically, it is known that high levels of agglutinating antibodies are protective [10]. Agglutinin response is related to FHA, lipo-oligosaccharide, PRN, and FIM2 and 3. Results from household studies in Sweden and Germany indicate that antibodies against PT and PRN correlate with protection [10,11,12]. Most likely, the protection is multifactorial and is related to both humoral and cellular immunity both in local mucosal surfaces and in systemic immune systems [13,14,15,16].

There is a two-phase decay of antibodies against PT, both after disease and after immunization. In Denmark, Dalby et al., showed that the biphasic decay, consisting of a rapid decay followed by a slower long-lasting decay, was faster after the disease (median half-life 221 days) than after adult booster vaccination with hydrogen peroxide inactivated pertussis vaccine (median half-life 508 days). The decay of antibodies after the disease was not age-related [17]. Hallander et al., (2005) studied the decay after the last dose of primary immunization in infants (2-, 4-, 6-month schedule) during the Swedish vaccine trials with chemically detoxified ACVs. The initial rapid decay occurred in 8–9 months, and in 65 months 50% of the children had undetectable antibodies against PT [18].

Denmark is the only country where protection against pertussis is nationally provided with a hydrogen peroxide inactivated single pertussis antigen vaccine, which was started in 1997. In 2003, a five-year booster was introduced. After 2005, the situation of pertussis seemed to be rather stable (population-level incidence < 10/100.000) until outbreaks occurred in 2012, 2016, and 2019 [19,20]. The incidence of pertussis in infants has been much higher than in the neighboring Nordic countries, and one reason is speculated to be the lack of adolescent boosters. However, it should be kept in mind that direct comparisons between countries with different surveillance systems for pertussis incidence are challenging [20,21]. Temporary maternal immunizations were introduced in Denmark in November 2019 after a nationwide epidemic started in July (*Statens Serum Newsletter*, No 42/43—2019). However, in 2019 Denmark also changed to an ACV vaccine containing PT and FHA (https://en.ssi.dk/news/news/2019/new-vaccine-formulation-in-the-childhood-vaccination-programme, accessed on 15 May 2021).

The problem with all pertussis vaccines has been the relatively rapid waning of immunity, especially against PT. It was also found that the immunogenicity of ACVs is higher in infants than in adolescents [16,18,22]. Since the enzymatic activity of native PT is detoxified for use in ACVs, most often chemically with formaldehyde, glutaraldehyde, or hydrogen peroxide, it has been proposed that the repeating use of ACVs induces B cells preferentially recognizing PT epitopes, which are induced by the chemical treatment of the toxin rather than epitopes against the native PT [23]. Thereafter, new, live attenuated bacterial vaccines were developed for intranasal use to mimic natural infection [24]. During active development, new vaccines with several different preventive strategies will be needed, and these should be tested in different age groups. The change to new vaccines is not going to be easy, since pertussis vaccines are one of the key components in the basic primary immunization of all children, globally.

A great number of studies are focused on the quantity of anti-PT IgG antibodies created after vaccination or infection. It is commonly accepted that a high quantity of anti-PT antibodies should give enough protection against the disease [11,25]. However, this approach does not describe the functionality of the antibodies. In order to provide more definitive evidence for vaccination- and infection-induced PT-specific antibodies’ ability to prevent the diverse biological activities of this protein, different methods have been developed in the field. These methods include, e.g., the neutralization of leukocytosis-promoting activity, antibody-mediated opsonophagocytosis [26], and importantly the measurement of the neutralization capacity of anti-PT antibodies, which can be evaluated by the antibodies’ ability to prevent the clustering of Chinese hamster ovary cells [27,28,29]. Other general aspects, such as the affinity of antibodies and their specific binding locations to PT, may act as contributive factors in the antibodies’ potency to neutralize and clear out the toxin [30,31]. In addition, it is crucial to understand the true nature of the machinery to produce these antibodies such as B cell populations, including memory B cells (Bmem), behind the antibody production and how they change during the lifespan [32,33]. Our aim in this review is to focus on functional antibodies to PT and their impact on improved understanding of anti-PT antibodies after vaccination and infection.

## 2. The Role of Neutralizing Antibodies to Pertussis Toxin (PTNAs)

Neutralization capacity of vaccine-induced and naturally boosted antibodies is commonly evaluated for virological or toxin-mediated bacterial studies to conclude good protection against the target pathogen [34,35,36,37,38]. In regard to pertussis studies, this approach has been used to study the quantity of toxin-neutralizing antibodies. However, it has not been successfully demonstrated that neutralizing antibodies would act as a correlate of protection against the disease in humans [39,40]. A commonly used approach to measure neutralizing capacity of anti-PT antibodies is based on the clustering effect and form changes of Chinese hamster ovary (CHO) epithelial cells. This method was originally used to study the activity of other AB_5_ toxins such as cholera toxin [41]. During the 1980s, the first CHO cell studies regarding PT were published, and they proved to be comparable to results from other toxins with similar CHO cell-clustering effects [27,28]. The method includes the following steps: (1) PT and serially diluted sera are first mixed together on a microwell plate; (2) the plate is incubated for 2–4 h; (3) a known number of CHO cells are added on the wells; and (4) the plate is monitored for up to 48 h for the formation of CHO cell clustering (Figure 2). The final neutralizing titer is notified in the well without any clusters [28,29].

The CHO cell assay has been widely used to study PT activity within pertussis vaccines to show that PT is properly detoxified [7,23,42,43]. Furthermore, it has also been used to show the protective properties of monoclonal antibodies (Mabs) targeted to different epitopes of PT subunits [5,44]. These epitope studies highly increased our knowledge of structure-based protection against PT and are described in more detail within the section of “PT-specific epitopes after infection and vaccination“.

However, none of the epitope-specific neutralization studies describe how the results were related to actual protection in humans acquired from vaccination or infection. Granström et al., and Trollfors et al., performed the first studies concerning this aspect. They showed a clear increase of the neutralizing titers (≥4-fold) after clinical infection, which remained high up to two years in patients that could be followed, despite a rapid initial decrease [45,46]. Soon after this, Granström et al., focused on serological correlates for protection against pertussis with pre-disease samples. According to their results, both antibody titers and neutralizing capacity indicated that anti-PT antibodies are crucial for protection [40]. Podda et al., focused on childhood vaccination and showed how PTNA titers increased from 1.1 (pre-vaccination) to 269.1 (after three doses). Interestingly, the titer increased highly after the second dose, whereas after the first dose there was almost no increase in the titer, an observation that also was shared in other studies [8,47]. These neutralizing antibodies were still detectable after five years [48]. Both chemically detoxified and genetically modified PT vaccines have demonstrated high post-vaccination neutralizing antibodies after one month, whereas WCVs induce a considerably weaker response, which was attributed across all studies to be closely related with the overall capability to induce high anti-PT IgG titers [23,49,50,51,52,53,54,55].

Another approach to study PTNAs was a PBMC-based method in which neutralization was monitored through the mitogenic activity of PT. These results were compared to the CHO cell assay. The investigators showed increased titers either post-vaccination or after infection compared to pre-vaccination sera. They also plotted anti-PT ELISA titers against CHO/PBMC assay titers, but the correlation was poor for both assays (r^2^ = 0.37/0.02, respectively). Therefore, the authors speculated that the ELISA, CHO, and PBMC assays measure distinct fractions of anti-PT antibodies [56]. In contrast, the majority of CHO-cell-based assays have shown a clear correlation (r > 0.80) between the concentration of anti-PT IgG antibodies and CHO cell neutralizing titers [46,47,48,49,55,57,58,59], concluding that basic ELISA measurements could be used to demonstrate the neutralization capacity of antibodies [48,57,59]. Furthermore, most of the neutralization and protection is based on anti-PT IgG antibodies, whereas IgA and IgM play a minor role [59]. Another novel study regarding PTNAs in infected subjects concluded a similar correlation between the quantity of anti-PT IgG antibodies and neutralizing titers (r = 0.68). A good correlation was found among those previously vaccinated (r = 0.71) subjects, but within the unvaccinated group with pertussis, a poor correlation (r = 0.26) was noticed. This also indicates good memory of the vaccine-based response [29]. Furthermore, a Japanese seroprevalence study among the heathy population classified the ratio between neutralizing titers and total IgG antibodies as low (<0.6), mediocre (0.6–<1.8), and high responses (≥1.8). The group of <0.6 ratio (cases with lower neutralization than expected based on anti-PT IgG) was the most common (over 60%, N = 106). Although the distribution of subjects into these three categories was similar between among groups, the correlations between CHO cell titers and anti-PT IgGs, decreased with age [60].

In the light of different studies performed, PTNAs do correlate with the total amount of anti-PT IgG antibodies and are induced considerably both after infection and vaccination, but they also indicate that high antibody concentration does not always guarantee good neutralization capacity [29,47,55,57]. Although these studies increase our understanding of PT antibody-based protection, further modifications for future testing might be needed to better show the actual fraction of neutralizing antibodies after vaccination or natural infection. Since the CHO cell assay is laborious, a high throughput assay should be developed in the future. Moreover, how to define a good neutralization capacity is another question to be considered. The CHO cell assay only measures the ability of antibodies to block the intrusion of PT into these cells. Thereafter, other abilities of the antibodies not covered in this review, e.g., to prevent leucocytosis, opsonophagocytosis and bactericidal capability are other critical lines of interest [26,61,62].

## 3. B Memory Cells as Carriers of Improved Antibodies

Bmem cells are essential for long-term protection against pertussis. Total populations of Bmem cells can be identified by flow cytometry, whereas antigen-specific responses need to be measured by ELISpots [63,64]. The B cell differentiation and maturation are highly dependent on the CD4+ T helper (Th) cells, especially follicular Th (Tfh) cells. Tfh cells guide B cells to germinal centers and orchestrate their differentiation into Bmem cells or long-lived plasma cells [32,65]. Notably, B cell populations do change with age, and a phenomenon called immunosenescence occurs. In general, this means that young individuals usually have a high number of naïve B cells, whereas Bmem specificity increases with age. However, if there is no natural or vaccine-based reoccurring boosting, elder individuals may lose some of the specific Bmem cells [33] and consequently affect the production of functional antibodies.

In general, the Bmem cell studies regarding pertussis are scarce. Still, the ones executed do offer a good overall view regarding specific Bmem patterns. Age-related differences in Bmem cell populations were studied within pertussis patients by van Twillert et al. Their results indicated that approximately 20% of the B cells in children were Bmem cells, whereas this fraction increased to 33% in adults. Although the quantity of anti-PT IgG antibodies decreases more rapidly than FHA and PRN, the decay of specific anti-PT Bmem cells was slower [66]. This further indicated that focusing only on the quantity of anti-PT IgG antibodies may skew the overall picture of B cell immunity against pertussis. Similar findings were shown in another Dutch study among different-aged subjects and different vaccination histories (no primary vaccination (born 1940–1957 N = 19) vs. primary vaccination (born 1957–1975 N = 17, young adults N = 24 and children N = 12)). Interestingly, the number of PT and FHA-specific Bmem cells was quite similar in each age group, although their vaccination background was different, and one can assume that unvaccinated subjects have received their protection from natural infection [67]. Two studies by Hendrikx et al., summarized findings from vaccination studies. Similar findings to van Twillert et al., were found among the total B cell populations within different age groups. Nevertheless, the pool of anti-PT Bmem cells was absent before ACV boosting, whereas Bmem cells against other antigens (PRN, FHA) were found [68,69]. However, after the ACV booster dose, PT-specific Bmem cells increased and were detectable at least after two years, although the number of these cells waned as fast as Bmems to FHA and PRN. The correlation between the quantity of anti-PT IgG antibodies and specific Bmem cells was better than the correlation for PRN and FHA. Furthermore, different vaccination background (ACV vs. WCV) also seems to influence the developing memory [13]. A study by Carollo et al., conducted among Italian children showed that the anti-PT Bmem cells are preserved even after five years of vaccination (more than 80% of the subjects had visible Bmem populations) with two different vaccines, although the responses were low (Geometric mean (GM) of PT-specific Bmem cells among the total population of Bmems was between 0.21–0.30). Remarkably, the differences of antibody quantities to FHA and PRN were not that high, although the number of antibodies was significantly lower for PT [14]. When we combine the findings from these studies, after primary vaccination PT-specific Bmems are preserved at least up to five years, although the frequency is low (GM: 0.3-0.4). However, booster doses highly increase these numbers, and after a few years increase can still be seen (GM: 1.9-3.2). Boosters also show an elevation in the specific Bmem quantity after one year of vaccination (GM: 8.3) [14,67,68,69].

It is well known that pertussis vaccines contain different concentrations of antigens, but usually, the amount of PT is one of the highest [22,25]. In this regard, it is very difficult to conclude why vaccinations do not induce as good B cell memory as infection to PT, although Bmem to FHA and PRN is rather high. Waning or even disappearing of anti-PT-specific Bmem cells has been speculated previously and one possibility for the imperfect memory could be insufficient stimulation of the antigen-specific B cells by vaccination or even natural infection [70]. Regarding the functional anti-PT IgG antibodies, Schure et al., also showed that a fraction of anti-PT IgG4 antibodies was highly upregulated one month post-vaccination, but for some reason these antibodies undergo a higher fold-decrease than other subclasses and were almost totally diminished after two years. The role of the IgG4 (similar to IgE) antibodies is still unclear, and they were shown to be created more by subjects primed by ACV vaccine when compared to WCV/natural infection [13,71]. The leading cause could be a Th2-type of response after repeated ACV vaccination, whereas the IgG2 subclass is more upregulated by the WCV vaccination [72]. How the Bmem cells are preserved to each different subclass remains to be studied.

The role of specific Bmem cells producing anti-PT IgA or IgM antibodies has also been studied among pertussis patients. Although the number of Bmem cells was relatively low, the ratio between specific anti-PT IgG and IgA Bmem cells after infection was defined around 2:1. Furthermore, there is also an increase in the number of IgM Bmem cells, but these seem to appear after any infection and specificity is hard to define. Still, the role of Bmem cells producing IgM should not be forgotten and further studies are needed, especially among vaccinated subjects [64]. Lately, studies done with an intranasal pertussis vaccine, inducing mucosal immunity, have shown good anti-PT IgG Bmem cell responses within the study subjects one month post-vaccination. However, most of these Bmem cells had already disappeared after 5–6 months [73]. A novel study performed with the same vaccine indicated that both anti-PT IgG and IgA Bmem cells were identified among the study subjects, although the response to IgA was lower [74]. The role of IgA Bmem cells on mucosal immunity has been speculated previously. According to this, there is no clear evidence that IgA antibodies are that crucial for the protection against pertussis based on mice tests, and so far no study performed with humans supports this either [75].

The Bmem cells do play an important role based on the findings from many studies, and particularly how they are boosted throughout the lifespan of an individual plays a major role in the protection. We still do not exactly know to which epitopes of the PT-specific memory is directed. Probably, some of the Bmem cells produce antibodies, which are created to certain fractions in the PT and which have gone through a formation change during the detoxification process. These Bmem cells are constantly boosted by vaccination and may not be protective. How to define this issue needs further studies and perhaps new approaches. Since the avidity of antibodies is due to the clonal expansion of Bmem cells, Bmem cells and the avidity of antibodies should be analyzed together.

## 4. Avidity as a Marker of Effective Antibodies to PT

Antibody avidity is referred to as the binding strength of the overall polyclonal antibody response, which may in itself reflect the efficiency of the circulating antibodies [76] and possible correlation for long-term [77,78,79] immunity against pertussis [30]. After exposure to PT, by vaccination or infection, the humoral immune response consists of the maturation of antibodies over time [80,81]. Avidity maturation evolves as an antigen-driven selection process [82,83] within the germinal centers [84,85], and as a result of somatic hypermutation [86] leads to the progressive selection of high-affinity antibodies [78,87,88,89]. Thus, avidity maturation may serve as a surrogate marker for memory priming and clonal selection of high-affinity Bmem cell clones [78,88].

Avidity is generally measured by ELISAs, in which the proportion of strong binding antibodies to PT is estimated after treating the antibodies with chaotropic solutions, which break down weak antigen–antibody bonds (Figure 3). Most often, a constant dilution or a titrated sample (serum but also plasma) is incubated, and the formed complexes are exposed to single or titrated concentrations of the chaotropic agent. This equilibrium is then compared to a state without a detergent (such as assay buffer). Alternatively, a fixed concentration of antibody is incubated with single or increasing concentrations of the detergent [90,91]. Wide ranges of detergents such as urea, ammonium thiocyanate, and diethylamine with alternating concentrations have been used in pertussis studies [30,60,89,90,92,93,94,95,96,97,98]. Avidity index (AI) is commonly used to characterize the functionality of antibodies based on the proportion of strong binding of antibodies to an antigen after treating the antibodies with chaotropic solutions. A variety of alternative models to present avidity have been used across studies ranging from the flat AI values, IC50-values (either based on detergent or serum dilution series) [90,96], AI thresholds (40–60%) [67], and combinatory models that aim to fractionate respective avidity proportions across detergent ranges or to account for overall antibody IgG titers (Table 1) [99,100]. Additionally, studies by Abu-Raya et al., present alternative calculation methods to give more emphasis on antibody avidity remaining at high detergent concentrations [99].

Most studies have focused on investigating avidity after ACV vaccination. These studies claim that avidity towards PT increases after vaccination [30,80,89], which is affected by the previous priming by either ACVs or WCVs [95], and that avidity declines alongside overall IgGs over time since primary vaccination [60]. Maternal vaccination with ACV was shown to increase the avidity of newborns’ anti-PT IgG, particularly if mothers were vaccinated in the early third trimester compared with the late third trimester [93,100], although this finding was not observed in a study by Maertens et al. [97]. In regard to infection, a study by Barkoff et al., demonstrated that a sharp increase of avidity between the acute and the convalescent-phase sera of culture-confirmed patients was observed, and a study by Hovingh et al., showed that the attained avidity remains high after the symptomatic phase up to three years after recovery [26,30]. Taking these observations together, anti-PT avidity is affected by the recency of vaccination, vaccine formulation, and exposure background (Knuutila et al., unpublished data). Although recent vaccination induces high avidity antibodies, the kinetics of the avidity responses between vaccination and infection seem different, highlighting a need for future vaccination research.

AI of PT antibodies was affected by detergent concentrations [80,99,100], antibody concentration, and assay parameters such as incubation time and temperature [80]. Together with the wide variety of starting approaches to methodology and possible interpretation models of data (Table 1), it becomes particularly challenging to make solid conclusions across studies on the field. Presumably, the most reliable assays would utilize both dilution series of samples and detergent; however, this is reagent- and time-consuming. The approach of detergent dilution series would be, at least for now, essential to visualize the whole spectrum of antibodies, as we yet lack the knowledge of a clinically relevant level of avidity for protection [15,99,101].

In short, to date studies regarding avidity are limited in number, and respective research settings have rarely been replicated. Therefore, a thorough overview concerning avidity across different populations would be needed. Moving forward, discussions about the unification of methodology would be helpful to limit the possibility of misinterpretation across studies. Importantly, the interplay of avidity among other antibody functions remains underreported.

## 5. PT Specific Epitopes after Infection and Vaccination

Because of PT’s toxic properties, the enzymatic activity of native PT is detoxified (PTd), for use in ACVs, most often chemically with formaldehyde, glutaraldehyde, or hydrogen peroxide. Alternatively, the enzymatic activity is inactivated through precise genetic modifications. The detoxification procedures inactivate the mitogenic, hemagglutinating, and ADP-ribosyltransferase activities of PT and stabilize the molecule, but the process also modifies the protein structure and surface epitopes. More specifically, formaldehyde treatment of PT inactivates the enzymatic activity of S1 (substitution of two amino acids Arg-9 →Lys and Gl-129→Gly) more effectively than glutaraldehyde, and conversely less in S2–S5 (Figure 1) [102,103,104,105]. According to solid-phase ELISA, genetic, hydrogen peroxide and 0.35% formaldehyde detoxification of PT resulted in reduced epitope binding in 2.9%, 31.4%, and 78.1% of the Mabs panel, respectively [102]. In the same study, the inactivation of PT with hydrogen peroxide also impaired the epitope binding activities of Mabs towards the molecule; however, fewer epitopes appeared to be affected in comparison to other chemical treatments [102]. Altogether, these treatments cause vaccine-induced antibodies to preferentially recognize PTd over native PT and could therefore affect the protective outcome through the quality of antibodies [30,102,106].

The connection between antibody-mediated neutralization of specific PT epitopes and protection from disease by human antibodies has not been studied extensively. Thus far, a deficit of vaccination-induced epitope-specific antibodies in comparison to infection was reported in studies by Sutherland et al., and Knuutila et al., by competitive ELISAs towards epitopes 1B7 and 1D7 in S1,11E6 in S23, and 7E10 in S3, which were identified with Mabs originally isolated from mouse [5,31,106,107,108]. After infection, antibodies binding the epitope recognized by the 7E10 Mab in particular appeared in substantial amounts and correlated with the overall anti-PT IgG titers, whereas children who received ACVs had much lower and non-correlative levels of 7E10-like antibodies regardless of dose amounts or detoxification treatment [31]. A similar observation regarding the 11E6 epitope was established in both studies. The presence of S1 epitopes was more vaccine-dose dependent: single doses of ACVs had fewer epitope-specific antibodies than infection, whereas three consecutive ACV doses in babies induced at least an equal amount of S1 antibodies as infection [31,106].

The functional protective characteristics of these Mabs were evaluated in the mouse model with both aerosol and intracerebral challenge models, successfully neutralizing a variety of PT functions. Furthermore, the two Mabs 1B7 and 11E6 were protective in baboon models, showing promise also as a therapeutic treatment for the disease [109,110,111]. Sutherland et al., found a correlation (Pearson r = 0.45) between human-induced 1B7-like antibodies and protective CHO-cell assays, which included samples from both vaccination and infection [106]. The epitope recognized by the 1B7 Mab remains the most studied epitope by far, and its location on PT has been defined in great detail [112,113,114]. Furthermore, the study by Acquaye-Seedah et al., found antibodies after ACVs from genetically distinct plasmablasts to 1B7, but those were not as potent to neutralize PT as the 1B7 reference Mab. Of interest, out of these clones, no clone was bound to the 11E6-like epitope [115].

Therefore, it seems evident that the current ACV formulations significantly affect the full capability of the epitope specificity of the formed antibodies and their effectiveness. Important epitopes already characterized in S1 and S2/3 within human infections, and in mouse and baboon models, suggest different inactivation strategies to blockade the receptor binding and catalytic pathways of PT, and represent potential serological correlates for next-generation vaccine development or even as biomarkers for separating infection- and vaccination-induced anti-PT antibodies.

Multiple studies have described PT-specific Mabs, antigenic determinants, and antibody recognition sites of PT from either large domains or peptide fragments. However, in contrast to other pathogens, recombinant truncated S1 molecules and synthetic peptides to PT have been studied in this regard rather little for pertussis [116]. No extensive epitope mapping exists today for antibody response in humans. For example, we have rather poor insight if ACVs skew the immune response to non-protective epitopes [31]. Likewise, WCV-induced epitope-specific antibody responses have not been studied [116]. Alternatively, genetically detoxified PTs have only two amino acid substitutions in the active site of the enzyme that result in the loss of ADP-ribosylation activity. They are safe and elicit an immunogenic response similar to the native PT antigen [23,49,117,118,119]. Therefore, genetically detoxified PT may also demonstrate a more similar presentation of anti-PT epitopes to infection as PTd, including vaccines, as it maintained both T- and B-cell epitopes [120]. Further characterization of PT epitopes, both from natural infections and vaccination, is needed to help better select PT-inactivation and stabilization methods, preserve potential protective vaccine epitopes, and better understand waning protective immunity to ACVs. In future work, epitope mapping, immune profiling, utilization of epitope databanks, and whole-genome sequencing are likely to be instrumental in determining whether protection against PT requires antibodies binding discrete epitopes versus a more diverse antibody response [121]. In addition, the constant evolution of *B. pertussis* may lead to key changes in PT epitopes [122,123,124,125] and therefore hinder the vaccination-induced protection. In addition, identified epitopes may be important for the development of Mabs, which could be used for the treatment and prevention of pertussis.

## 6. Final Conclusions and Future Perspectives

The recurrent pertussis epidemics in immunized populations have shown that blind faith in the quantity of anti-PT antibodies does not give us a whole picture of long-lasting protection. To evaluate effective antibody response after vaccination and infection, both quantity and quality of anti-PT antibodies should be determined. PTNAs and anti-PT antibodies with proper avidity are essential aspects to consider for good protection against the disease and infection. Although the CHO cell assay reveals how antibodies prevent attachment of PT to cells, the neutralizing capacity to prevent lymphocytosis and diminishing the bacterial load, as shown in the baboon studies, should also be a topic of interest for future studies. Furthermore, although the scale of recognized PT epitopes in studies has been wide, our knowledge of the human-induced different epitopes is still very narrow, with a main focus on only two epitopes (recognized by Mabs 1B7 (S1) and 11E6 (S23)), especially when compared to key antigens of other diseases. PT-specific Bmem cells seem to last at least five years after vaccination, but we still lack the information on what happens after this time period. In addition, the effect of vaccination background on the responses should be taken into account in further studies and is needed for vaccines under development.

Remarkably, no correlate of protection against pertussis has been found, although the many aspects have been considered and studied. In addition, a clear indication of what titer of neutralization is effective in vivo is lacking. Likewise, an overall image for a sufficient level of PT-based protection after infection is needed, as whether a long-lasting quantity of (neutralizing) antibodies counts more than quality through other means such as epitope specificity and avidity, or whether the interplay of all factors has to be considered. Thereafter, new approaches to determine the functionality of antibodies should be investigated. New vaccination strategies are needed, whereas boosting ACV-primed adolescents with recombinant genetically detoxified PT vaccine could be one way to go as it induces higher anti-PT and PT-neutralizing responses than chemically detoxified ACV and further increased PT-specific memory B cells in these individuals [23]. However, despite this superior immunogenicity, the recombinant vaccine may have to be given repeatedly, earlier, and/or with novel adjuvants. Furthermore, the nasal vaccines containing either live attenuated bacteria or outer membrane vesicles are also intriguing solutions, as the results so far look promising [126,127]. Nevertheless, we need more information regarding subjects infected with pertussis (both with classic and mild pertussis). A currently ongoing PeriScope project is aiming in this direction with a human challenge model and with household studies [128]. Similar studies to PeriScope with a deep focus on quantity and quality of anti-PT antibodies are further needed in the future to determine the correlate of protection for pertussis, although other pertussis adhesins and toxins should not be forgotten either. Based on the present knowledge, and because of the unique character of PT for *B. pertussis*, the measurement of vaccination- and infection-induced anti-PT antibodies to qualitative serological determinants will be crucial in the development of future vaccines to match long-term protection.

## Figures and Tables

**Figure 1 toxins-13-00508-f001:**
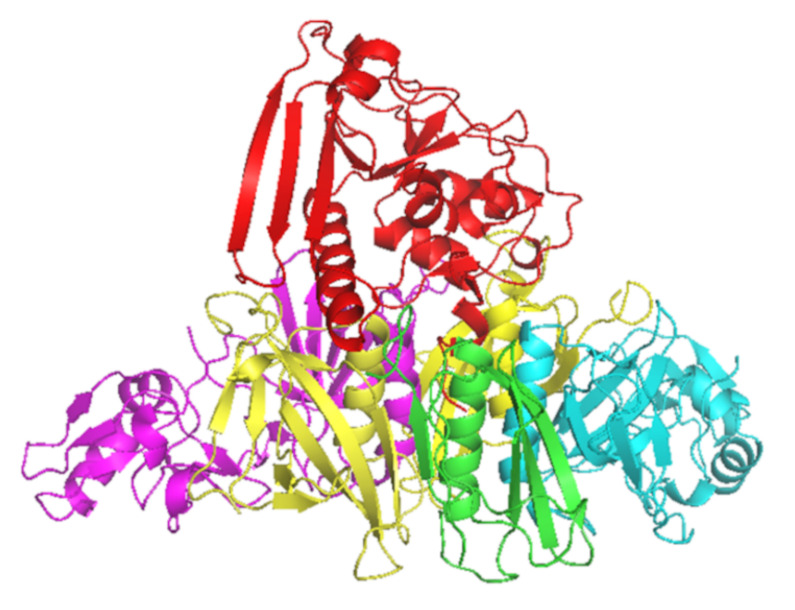
Crystal structure of PT subunits 1 (red), 2 (purple), 3 (cyan), 4 (yellow) and 5 (green). Generated by PyMOL (version 2.3.1, Schrödinger, LLC).

**Figure 2 toxins-13-00508-f002:**
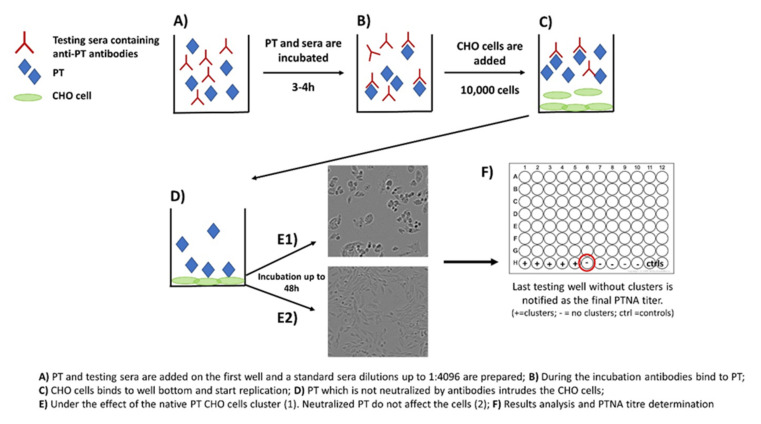
Principle of the CHO cell-based assay for neutralizing antibodies to pertussis toxin.

**Figure 3 toxins-13-00508-f003:**
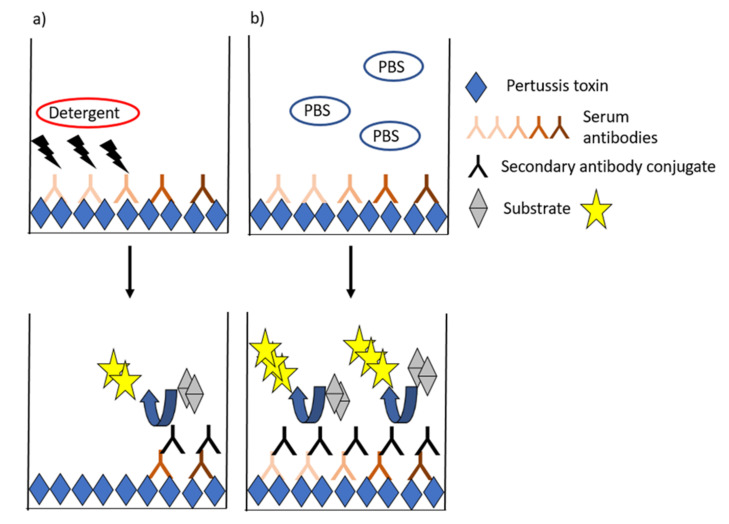
General ELISA schematic for avidity assays. The presence of a chaotropic detergent (**a**) interferes with weak-binding antibodies, and the proportion of high-binding antibodies left at the binding surface is compared with a control state with only assay buffer (**b**).

**Table 1 toxins-13-00508-t001:** Common analysis models to PT avidity data.

Model.	Description	Advantages	Requirements
AI	The absorbance of detergent treated sample/Absorbance of the control sample	Simple	Only duplicate wells to be tested
AI thresholds	Avidity is considered strong if it exceeds a certain level, or weak if it does not	Simple to categorize strong and weak binding	Clinically or experientially set relevant thresholds to determine strong and weak binding
Fractional avidity	The percentage from overall avidity within a concentration range of detergent	The proportion of strong binding can be more accurately evaluated	Dilution series of detergent
IC50	Antibody or detergent concentration corresponding to 50% avidity	Less reliant on initial antibody or single detergent levels	Dilution series of sample or detergent

## Data Availability

Data sharing not applicable.

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
