# Peer review of "Evaluation of Anti-PT Antibody Response after Pertussis Vaccination and Infection: The Importance of Both Quantity and Quality"

_toxins, 2021, doi:10.3390/toxins13080508_

Round 1

Reviewer 1 Report

The statement that recombinant PT containing vaccine is more protective is an opinion, the facts are that in a head-to-head efficacy trial

chemically treated or recombinant PT showed equal short and long duration protection

It would have been good to describe the relative poor and short duration of protection after teenager boosting with aP vaccines

Comparing countries for pertussis incidence is difficult since surveillance systems differ by country

The opinion that in bacteriological studies there is only focus on neutralizing anti-toxin antibodies is incorrect, for may bacterial vaccines bactericidal and opsonophagocytic antibodies are

critical measurements

To try to use avidity as a measurement of inducing memory is attractive and showed some relevance for polysaccharide conjugate vaccines since polysaccharides

only induce T-cell dependent B-cell memory as well as antibody affinity maturation when presented as a conjugate. For protein based vaccines such surrogate measurements are not obvious

Reviewer 2 Report

Overall a worthy effort to describe assays/data/gaps on the immunological parameters that can be applied to study the responses to PT in a meaningful manner. 

Suggest to significantly revise the introduction to be more focused on providing precisely and concisely the details necessary to introduce the other sections and the objectives of the review. The reader may be naive or novice to the subject matter. E.g. the reader could be missing information on PT (e.g. secreted toxin) and the different forms of PT used in current or past vaccines versus native PT. Some of this is caputured in section 5 but the reader should be informed earlier.

Line 62 mentions an agglutinin response without FHA, while line 60/61 mentions FHA as a hemagglutinin.

Line 66 misses a citation. There are many recent papers that suggest cellular immune responses are important.

Line 68 - biphasic kinetics, it might be good to actually describe this kinetic in this paragraph.

Line 93 make sure the reader can understand that the new vaccine is a live attenuated bacterial solution, not another form of ACV. 

Line 113, I think the point of the authors is that for bacteria often the quantity of neutralizing antibodies was not demonstrated to be a correlate of protection? Provide more clarity for the statement.

Line 132 related, not reflected

Lines 161 following. The description and relevance of the japanese study is difficult to follow. The best correlation was found in the group with low PTNA and high IgG?  What about the comment in line 170 (high Ab conc. does not always guarantee good neutralization capacity)?

Line 181 is their much evidence that anti-PT would be implicated in preventing bacterial replication?

Line 279 following, include the principle mechanism of the assay into the body of text, not solely in the figure.

Line 309 observed, not notified

Line 310 include how many years

Line 336 following needs clarification. The genetically detoxified PT (R9K E129G) is only very midly treated with aldehyde, the enzymatic activity is removed by the mutations. I think the authors confuse the mechnisms of detoxification.

Line 360 as not with

Line 362 intranasal (?) not intracellular

Line 365 what is a decent correlation?

Line 375 more clarity "suggest different inactivation strategies to the different pathogenic mechanisms of PT"

Line 389 as not than

Line 426 what is the recombinant PT vaccine? Is this genetically detoxified? Clarify

Line 218 seems to be in contradiction to line 90 following where boosting with ACV is mentioned as eliciting epitopes directed against the chemically treated PT and not the native PT. How is this repeated boosting than beneficial? The reader not familiar with the topic of ACV may find this confusing. Please provide more clarity

Round 2

Reviewer 1 Report

I am OK with the adapted manuscript